# On the Uniaxial Compression Testing of Metallic Alloys at High Strain Rates: An Assessment of DEFORM-3D Simulation

**Michael Bodunrin [1,2,*]**, **Japheth Obiko [3]** and **Desmond Klenam [1]**

1. School of Chemical and Metallurgical Engineering, DSI-NRF Centre of Excellence in Strong Materials, University of the Witwatersrand, Johannesburg, Private Bag 3, Johannesburg 2050, South Africa
2. African Academy of Sciences, Nairobi P.O. Box 24916-00502, Kenya
3. Department of Mining, Materials and Petroleum Engineering, Jomo Kenyatta University of Agricultural Technology, Nairobi P.O. Box 62000-00200, Kenya
* Correspondence: michael.bodunrin@wits.ac.za; Tel.: +254-27117177506

**Featured Application: Metalworking.**

**Abstract:** In this study, the challenges associated with conducting high-strain rate uniaxial compression testing experiments are highlighted. To address these challenges, DEFORM-3D simulation was explored as an alternative approach to experimental testing. Previously established constitutive constants obtained from experimental low strain rate uniaxial compression testing of three titanium ($\alpha$ + $\beta$) alloys were used as input codes. From the results, the peak flow stress values obtained from the DEFORM-3D simulation were close to the values obtained experimentally at low (0.1 to 10/s) and high (20 and 50/s) strain rates. For the alloys considered in this study, a discrepancy of ~20% in the peak flow stress was obtained at a 10/s strain rate. The difference in peak flow stress for strain rates less than 10/s or higher (20 and 50/s) is within acceptable limits. The limitations of using DEFORM-3D simulations for high strain rate uniaxial compression testing are highlighted.

**Keywords:** flow stress; DEFORM-3D; hot working; Gleeble 3500; finite element method

## 1. Introduction

Thermomechanical processing, used interchangeably in this paper as hot working, hot forming, or hot deformation, has remained one of the oldest and most prominent processing techniques for shaping semi-finished or finished metallic components for a wide range of applications [1]. Rolling, forging, extrusion, wire drawing, and sheet metal forming are prominent examples of thermomechanical processing techniques [2]. Apart from component forming, the most important advantage of thermomechanical processing is its opportunity for microstructural control [1,2]. Hence, metallurgists can manipulate processing variables to achieve targeted microstructural features required to improve bulk and surface properties for optimal in-service performance [2].

Depending on the thermomechanical processing technique used, a wide range of process parameters are involved, but the key parameters that determine the response of materials to thermomechanical processing are strain rate, strain, deformation temperatures, and initial microstructure [1,2]. Flow stress, the primary response of metallic alloys to imposed hot working parameters, is presented as true stress-true strain data [3]. Consequently, analysis of the trends of the true stress-true strain curves was the foremost method used to understand the hot working behavior of metallic alloys. However, this approach is only suggestive and cannot provide a convincing explanation of the physical mechanisms controlling hot working processes [4].

The secondary method that follows includes constitutive analysis or processing maps [2]. The constitutive analysis of flow stress follows either a phenomenological

or physical approach depending on the prior information and the objective of the analysis [5]. For example, phenomenological constitutive analysis requires fewer constants and can be used to predict flow stress. However, it is not very effective in describing the mechanisms controlling the hot working process [6]. Although some authors attempted to apply some physical meaning to the phenomenological equation, which was successfully implemented on steel [6,7], the modified model could not effectively describe the flow behavior of titanium alloys, especially those with a complex initial microstructure. Lin et al. [5] comprehensively reviewed the different constitutive flow stress models. Physical constitutive models are quite reliable in predicting flow stress and explaining the physical mechanisms driving deformation processes, but they require prior knowledge of the material and lots of physical constants must be derived for them to be effective [5–7]. Therefore, using physical constitutive models to describe the flow stress for newly developed metallic alloys might be challenging. Additionally, the complexity of physical models has made them unattractive to researchers and experts in the metalworking sector.

The processing map, which started in its simplistic form as the Raj map, is now based on dynamic materials models and extremum principles of irreversible thermodynamics [2,8]. Researchers use this map extensively to identify dominant mechanisms driving deformation processes and optimize the hot working process. Although it cannot be used to predict flow stress, its versatility in establishing the optimum combination of hot working parameters for microstructural control has been well exploited in the metalworking industry. One key limitation of processing maps is the need for microstructural validation of its prediction because the instability criterion, based mainly on the extremum principle of irreversible thermodynamics, is not always accurate. Despite Murty et al.'s [9] recent modification of the instability criterion, microstructural validations of the optimum and unsafe deformation regions are still required.

It is worth mentioning that the validity of the secondary method heavily depends on the accuracy of the flow stress obtained from experimental testing. For this reason, parametric relationships are used to correct flow stress for adiabatic heating and frictional effects [10]. Frictional effects during hot working are carefully controlled using combinations of lubricants during experimental setup [11], while adiabatic heating may be controlled by selecting low strain rates that allow enough time for heat dissipation. However, this is not always practicable under industrial conditions like forging and extrusion, which could require very high strain rates. Nevertheless, the experimental setup plays a significant role in obtaining accurate flow stress data.

Uniaxial compression testing is one of the most utilized experimental testing methods for determining flow stress. The test is conducted under different conditions of strain, strain rates, and deformation temperatures. The common testing facilities include screw-driven machines, servo-hydraulic machines, and split Hopkinson pressure bars (SHPBs). The conventional screw-driven and servo-hydraulic machines like the Instron universal testing machine and Gleeble 1500 thermomechanical simulator are suitable for conducting tests up to a strain rate of 5/s, whereas specifically designed testing machines with higher servo-hydraulic and fast data acquisition systems like the Gleeble 3500 and Gleeble 3800 achieve strain rates of about 100/s [12]. However, these strain rates are much lower than those of typical industrial processes like forging. The SHPB can achieve higher strain rates between $10^2$ and $10^4$/s, which covers a wide range of industrial processing conditions but is limited to a max strain of 0.3 [12,13].

Given the varying capacity of the different testing machines, it is quite challenging to test a wider range of parameters using a single machine, which affects the reliability and validity of the flow stress data. The existing machines are based on slightly different principles and data acquisition systems, which may influence the results. Testing metallic alloys for a broader range of strain rates involves using two different machines [14,15]. Even in situations where a single machine is used, as in the case of Gleeble 3500 or 3800 thermomechanical simulators, testing at strain rates above 10/s not only results in frequent wear and tear of the anvils and wastes time, test samples, and consumables, but it

also gives jerky flow stress patterns that are not easily reproducible, and their origin can be misleading. These jerky flow stress patterns could result from metallurgical phenomena or vibrating load cells in the machine.

These challenges associated with high strain rate uniaxial compression testing led the authors to explore an alternative method to generate flow stress data. This method involves using constitutive parameters obtained from low strain rates in uniaxial compression testing as an input code in DEFORM-3D. Thereafter, DEFORM-3D numerical simulation was used to generate flow stress data under high strain rate uniaxial compression conditions. The method's reliability was then evaluated by comparing the generated flow stress data with experimental data.

## 2. Materials and Methods

### 2.1. Finite Element Simulation

The finite element method (FEM) simulation provides an accurate flow stress analysis and a quicker method to optimize the process parameters during forming. Hence, reducing the production cost and time [15]. The metal flow pattern obtained from the simulation analysis is of great advantage in the manufacture of high-quality products. In this study, DEFORM-3D software was used to investigate the influence of forming parameters on the flow stress behavior of titanium alloys (Ti-4.5Al-1V-3Fe, Ti-6Al-1V-3Fe, and Ti-6Al-4V). The general relationship between these process parameters (strain-$\varepsilon$, strain rates-$\dot{\varepsilon}$ and temperature-T) and flow stress ($\sigma$) is as given in Equation (1) [16].

$$\overline{\sigma} = \overline{\sigma}\left(\overline{\varepsilon},\ \dot{\varepsilon},\ T\right) \tag{1}$$

The flow behavior of titanium alloys was studied by inputting the constitutive parameters derived from experimental flow stress data [17–20]. The constitutive parameters used in this analysis are given in Table 1. The software input window is shown in Figure 1. The FEM simulation analysis uses a rigid viscoplastic formulation approach [16]. The governing equations of the metal-forming process are widely covered in the literature [16]. The die and workpiece design was carried out using the in-built software modeling module. The workpiece dimensions were similar to those used in the physical laboratory simulation. Table 2 shows the simulation parameters and conditions. The sample reduction in height was from 12 to 6.3 mm, $\varepsilon \approx 0.64$ for the low strain rate (0.1 to 10/s) simulations. A total strain of ~1.4 was used for the high strain rate (20 to 10,000/s) simulation.

**Table 1.** Constitutive parameters for titanium alloys investigated [20].

| Ti Alloy | $\alpha$ | $\beta$ | $n'$ | $n$ | $Q$ (kJ/mol) | $A$ |
|---|---|---|---|---|---|---|
| Ti-4.5Al-1V-3Fe | 0.008 | 0.04 | 4.958 | 3.335 | 465 | $1.20 \times 10^{20}$ |
| Ti-6Al-1V-3Fe | 0.006 | 0.03 | 5.595 | 3.808 | 487 | $1.50 \times 10^{21}$ |
| Ti-6Al-4V | 0.005 | 0.04 | 7.569 | 5.057 | 620 | $1.44 \times 10^{28}$ |

### 2.2. Experimental Validation

The DEFORM-3D simulation results were validated for low and high strain rate conditions. For the low strain rates (0.1 to 10/s), the peak stress from published flow stress data for three titanium ($\alpha + \beta$) alloys—Ti-4.5Al-1V-3Fe, Ti-6Al-1V-3Fe, and Ti-6Al-4V—were used [16–19]. The data were generated from uniaxial compression testing on the Gleeble 3500 thermomechanical simulator at deformation temperatures of 800 to 950 °C. For the high strain rate test, uniaxial compression testing was conducted on commercial-grade Ti-6Al-4V alloy with a complex initial microstructure. The complex microstructure consisted of nearly equiaxed alpha grains (diameter = 5 $\pm$ 2 μm), elongated alpha grains (length between 12 and 49 μm), and a network of intergranular beta phase. The test samples were 8 mm in diameter and 12 mm in length. Uniaxial compression testing conducted on the same machine was carried out at a deformation temperature of 900 °C, total strain of 0.6,

and strain rates of 20 and 50/s. The 900 °C deformation temperature was chosen since it was previously reported as a safe deformation temperature for the three titanium alloys [20]. The samples were heated directly to the deformation temperature and then held for 180 s prior to deformation for homogenization, and thereafter deformed at specified strain rates. A chromel-alumel (type K) thermocouple was spot-welded to the midspan of the samples to measure the temperature. Graphite foil and nickel paste were placed between the anvils and the samples to minimize frictional effects on flow stress. After deformation, the flow stress of the deformed samples was corrected for frictional effects following Obiko [10].

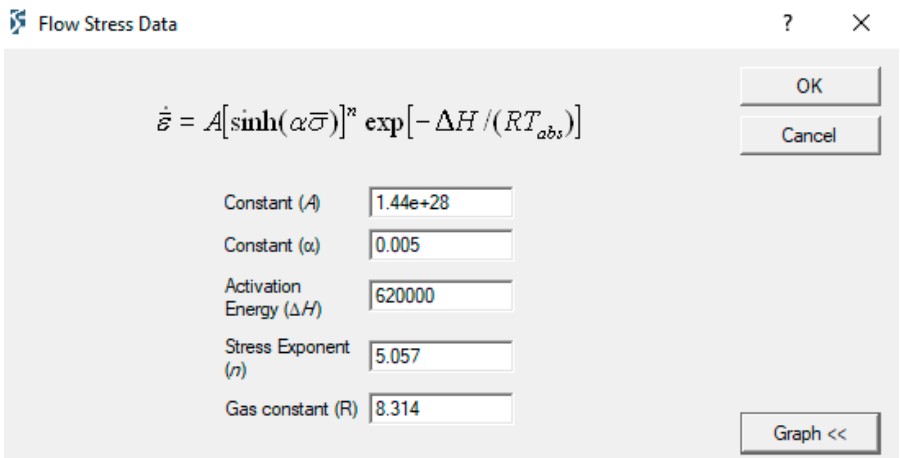

**Figure 1.** Deform-3D interface for computing constitutive constants.

**Table 2.** Forging simulation parameters using DEFORM-3D.

| Number | Description | Value |
|---|---|---|
| 1 | Workpiece material | Titanium alloys |
| 2 | Workpiece height (mm) | 12 |
| 3 | Workpiece diameter (mm) | 8 |
| 4 | Strain rate (/s) | 0.1, 1, 10, 15, 20, 50 |
| 5 | Workpiece temperature (°C) | 800, 850, 900, 950 |
| 6 | Initial die temperature (°C) | 200 |
| 7 | Coefficient of heat convection (N/(s.mm.°C)) | 0.02 |
| 8 | Coefficient of heat transfer (N/(s.mm. °C)) | 5 |
| 9 | Coefficient of friction (lubricated) | 0.3 |
| 10 | Finite element method elements | 26,247 |
| 11 | Element nodes | 5352 |
| 12 | Simulation environment (°C) | 25 |
| 13 | Friction type | shear |

The experimentally determined peak flow stress was compared with the DEFORM-3D simulation peak stress. Optical microscopy was performed on the deformed samples following standard metallographic procedures described by Bodunrin et al. [21].

## 3. Results

### 3.1. Flow Stress Prediction Using DEFORM-3D Simulation

Figure 2 compares the peak stress obtained from experimental uniaxial compression testing and DEFORM-3D simulation for three titanium ($\alpha + \beta$) alloys. The peak stress values are very close except for the simulation performed at 10/s for the experimental Ti-4.5Al-1V-3Fe and Ti-6Al-1V-3Fe alloys and the commercial Ti-6Al-4V alloy, where there

were observable deviations. This was expected because the simulations may not always accurately represent all the experimental testing conditions [22,23]. For example, a fixed coefficient of friction of 0.3 was used in the simulations for simplicity, but it varies rapidly under certain experimental conditions, affecting some simulation outputs like flow stress. Despite this deviation, Figure 3 shows a good correlation between the imposed deformation parameters and peak flow stress obtained from the FEM simulation. There was a high correlation coefficient ($R^2 > 0.98$) between peak flow stress and the strain rates for different test temperatures, even when the stimulation was extended from a 10/s to a 1000/s strain rate. This means that the constitutive parameters used to determine the peak flow stress during numerical simulation effectively describe the flow behavior at strain rates that are extremely higher than those tested in the laboratory.

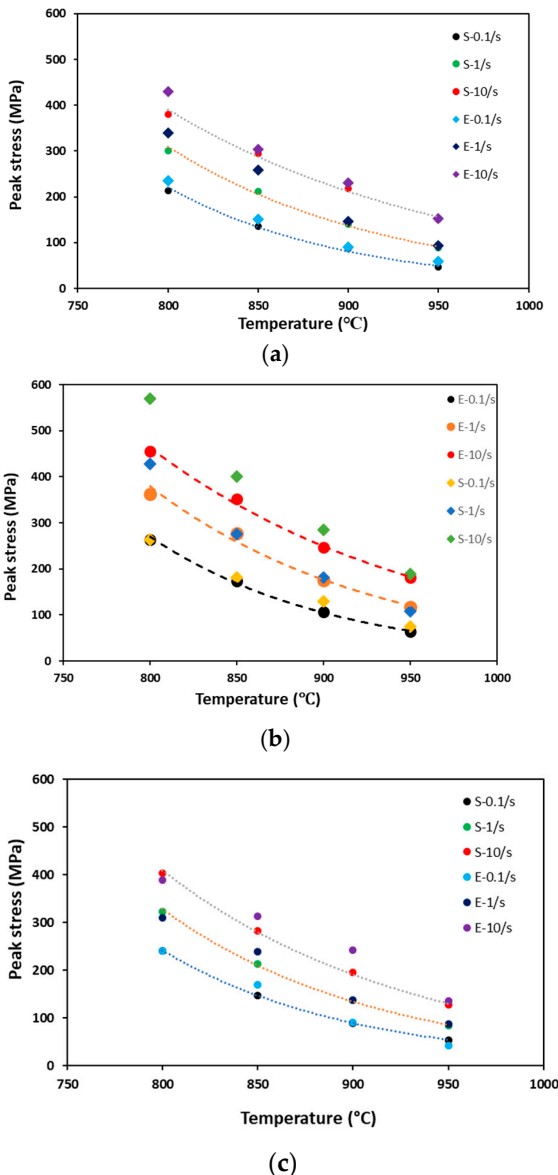

**Figure 2.** Peak stress obtained from the experiment and DEFORM-3D simulation (**a**) Ti-4.5Al-1V-3Fe; (**b**) Ti-6Al-1V-3Fe; and (**c**) Ti-6Al-4V. *S represents simulated peak stress, and E represents experimental peak stress.*



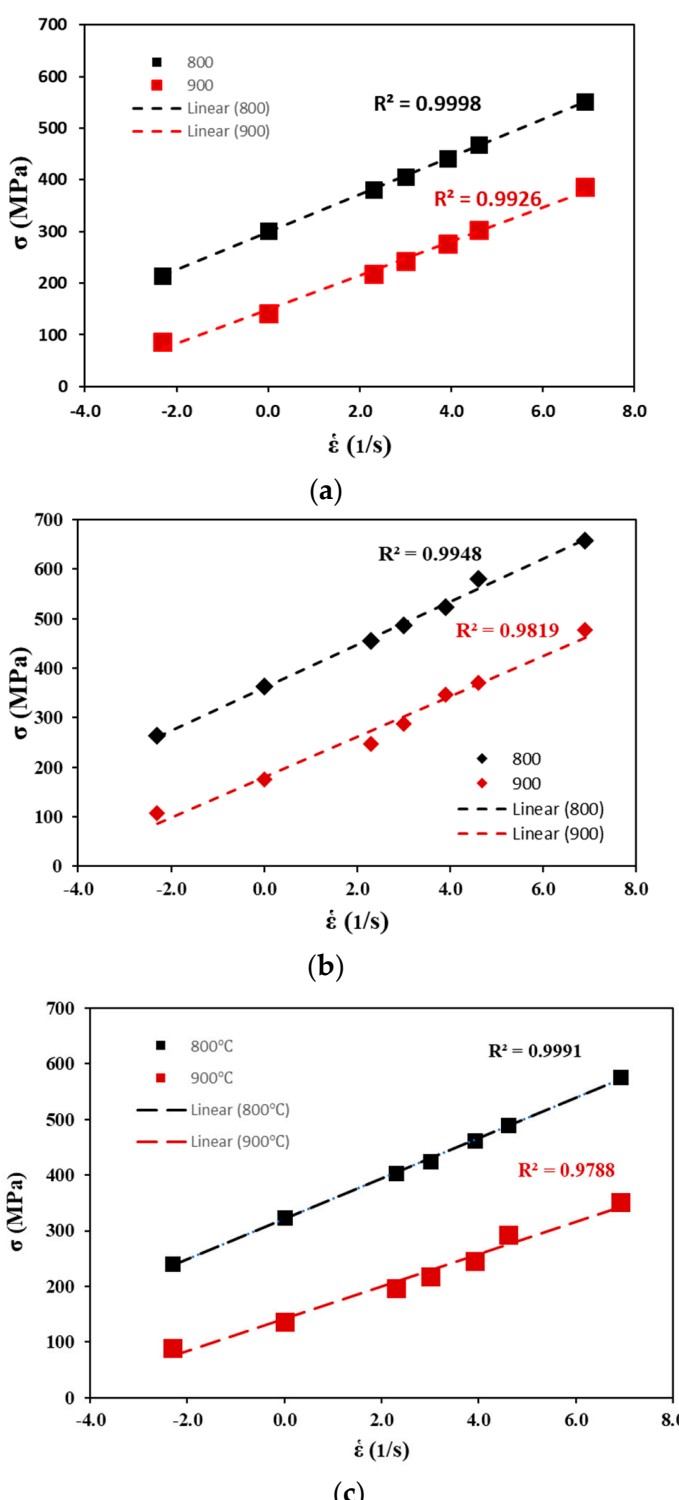

**Figure 3.** Simulated peak stress vs. linear strain rate at 800 and 900 °C for (**a**) Ti-4.5Al-1V-3Fe; (**b**) Ti-6Al-1V-3Fe; and (**c**) Ti-6Al-4V.

### 3.2. Experimental Validation of Flow Stress Predicted by DEFORM-3D Simulation

The flow stress obtained from uniaxial compression testing at high strain rates of 20 to 1000/s at two different temperatures for the three different titanium alloys is shown in Figure 4. As expected in materials with positive strain rate sensitivity, flow stress increased with increasing strain rate and decreasing deformation temperatures [17–20]. The flow curves show continuous flow softening after the peak stress was attained. This suggests

either geometric dynamic recrystallization of elongated grains or deformation-induced defects like cracking as the possible softening mechanism [17,18,20]. To confirm this trend, limited experimental uniaxial compression tests were performed at a deformation temperature of 900 °C and two strain rates (20 and 50/s) on the commercial-grade Ti-6Al-4V alloy. The corresponding optical micrographs of post-deformed samples were analyzed (Figure 5). The stress-strain curve shows continuous flow softening accompanied by flow oscillation. Flow oscillation is not observed in the DEFORM-3D-generated data in Figure 4, which could be because only the peak stress values were used to determine the simulation's constitutive constants. Should the constants be derived at incremental strain, the oscillations obtained experimentally may have been seen in the simulation results. Murillo-Marrodán et al. [24] reported that using constitutive constants obtained at 0.35 strain intervals from a physical model improved the flow stress generated by the FEM. The corresponding optical images (Figure 5b,c) show uniformly refined grains (diameter is ~12 μm) compared with preform microstructures in previous work [17,20]. The uniformly refined grains confirmed the dynamic recrystallization of the complex initial microstructure, particularly the elongated α-Ti grains during deformation. These features support the flow oscillations seen at 20 and 50/s, and the absence of deformation-induced defects indicates that the continuous softening was due to geometric dynamic recrystallization of the elongated α-Ti grains. Extensive details on the microstructural evolution during the deformation of the Ti-6Al-4V alloy at strain rates of 0.1 to 10/s have been reported previously [20].

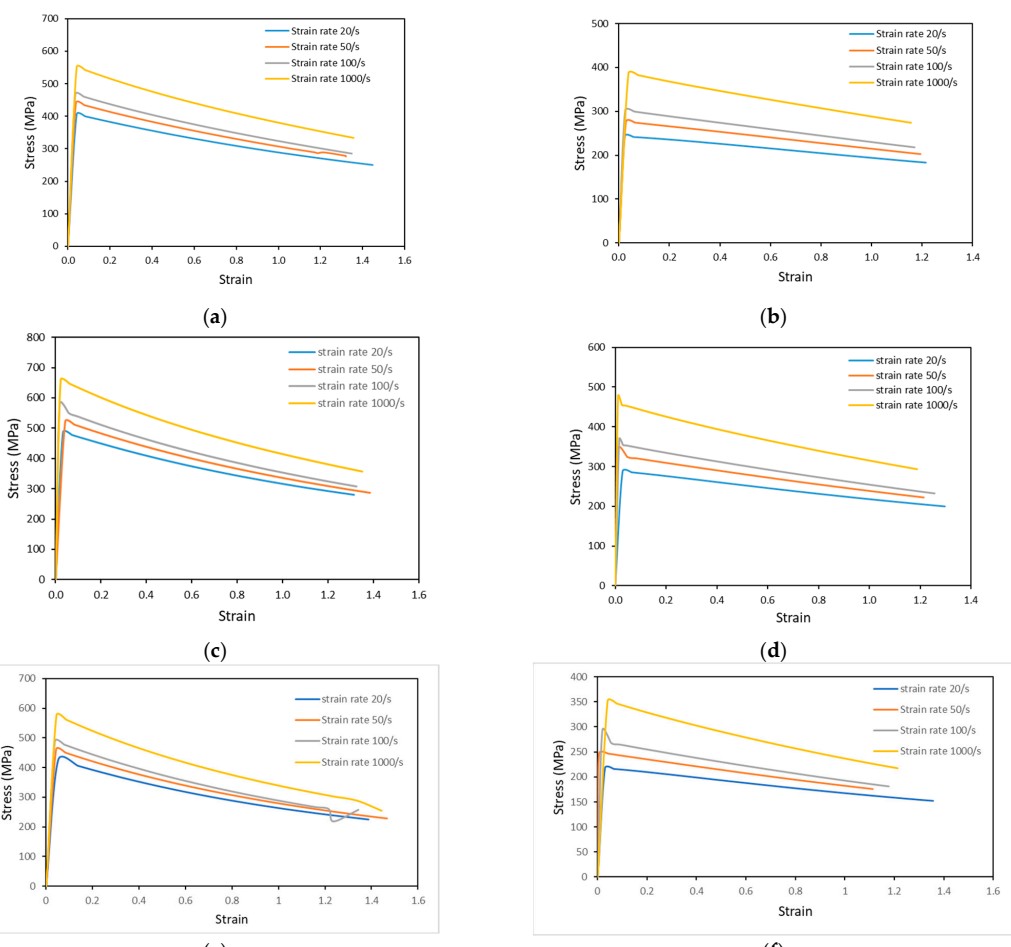

**Figure 4.** Flow stress obtained from DEFORM-3D simulation at high strain rates for (**a**) Ti-4.5Al-1V-3Fe at 800 °C; (**b**) Ti-4.5Al-1V-3Fe at 900 °C; (**c**) Ti-6Al-1V-3Fe at 800 °C; (**d**) Ti-6Al-1V-3Fe at 900 °C; (**e**) Ti-6Al-4V at 800 °C; and (**f**) Ti-6Al-4V at 900 °C.

To validate the simulation results, the peak stress obtained from the Gleeble experiment and that of the DEFORM-3D simulation were compared, and the peak stress values were similar. The difference in the peak flow stress values was 15% for a strain rate of 20/s and 6% for a 50/s strain rate, respectively (Table 3). This difference is lower than the ~19% obtained for a much lower strain rate of 10/s. These trends indicate that, in the absence of SHPB, high strain rate uniaxial compression data can be obtained from DEFORM-3D simulation in as much as the constitutive constants have been established from experimental data at lower strain rates.

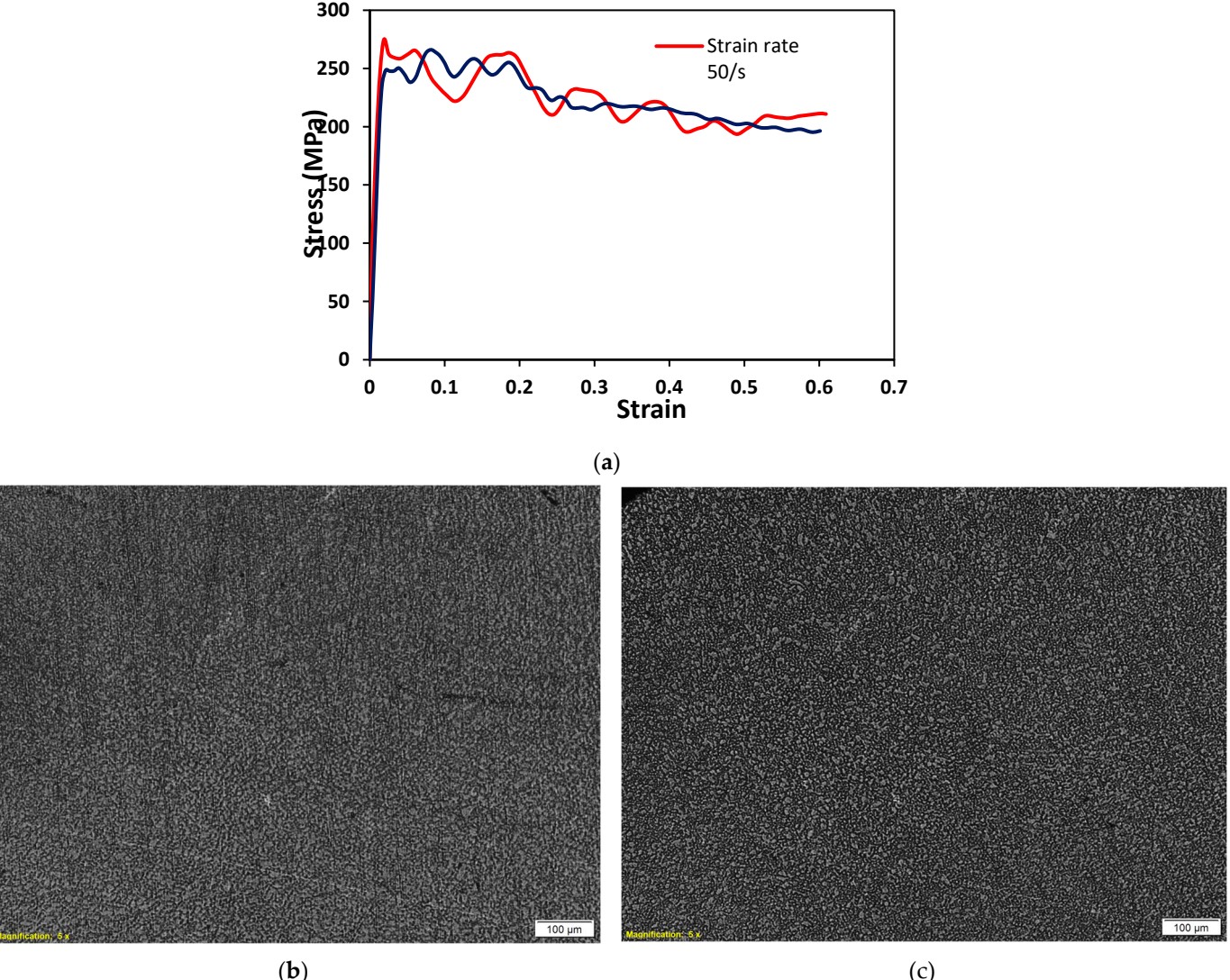

**Figure 5.** (**a**) The uncorrected flow stress of commercial Ti-6Al-4V obtained from the Gleeble 3500 uni-axial compression experiment and corresponding optical images at (**b**) 20/s; and (**c**) 50/s strain rates.

**Table 3.** Peak flow stress obtained under DEFORM-3D simulation and the Gleeble experiment for Ti-6Al-4V.

| Strain Rate (1/s) | Ln Strain Rate (1/s) | Simulated Peak Stress | Experimental Peak Stress * | Difference (%) |
|---|---|---|---|---|
| | | 900 °C | 900 °C | |
| 0.1 | −2.302585 | 89.8 | 88 | 2.05 |
| 1 | 0 | 137.1 | 138 | 0.65 |
| 10 | 2.3025851 | 196 | 243 | 19.3 |
| 20 | 2.9957323 | 217.6 | 247.8 | 12.2 |
| 50 | 3.9120230 | 246.3 | 262.7 | 6.2 |
| 100 | 4.6051702 | 293.3 | - | - |
| 1000 | 6.9077553 | 351.8 | - | - |

* Experimental peak stress was corrected for friction.

## 4. Discussion

The main question addressed in this study is whether DEFORM-3D simulation based on constitutive parameters derived experimentally from low strain rate uniaxial compression testing can be used to generate accurate flow stress data for high strain rate test conditions. In this study, we show that DEFORM-3D uniaxial compression simulation can be used to determine the flow stress that is comparable with what is obtainable experimentally, even at very high strain rates that are difficult to test in the laboratory.

The findings from this work are situated within the context of the challenges associated with conducting high strain rate uniaxial compression testing in the laboratory. There is currently no single piece of equipment that can be used to carry out compression testing over a wide range of strain rates that represent industrial hot working processes. Table 4 shows typical strain rates for different metalworking processes, which are between 0.0001/s for sheet forming and 1,000,000/s for high-speed machining. The screw-driven and servo-hydraulic systems have a limit of 5 to 200/s [12,13], which is far below the upper limit of 1000/s used in forging and rolling operations. The alternative, the SHPB, cannot be used at strain rates below 100/s [12,13]. Most hot-working laboratories are equipped with either screw-driven or servo-hydraulic systems, while few laboratories have an SHPB machine. Consequently, most studies on hot compression testing used a maximum strain rate of 10/s. In studies where a strain rate above 10/s was used, the results were not easily reproducible, and a great deal of time and resources went into conducting these experiments. Depending on the machine used, numerous parametric adjustments must be made based on trial and error to achieve the desired strain rates. This makes it difficult to have a general overview of the response of both commercial-grade and emerging metallic alloys over a wide range of imposed strain rates that cut across different industrial working processes.

In this study, we showed that established constitutive parameters from lower strain rate (0.1 to 10/s) uniaxial compression testing of three types of titanium alloys (one commercial-grade, Ti-6Al-4V, and two experimental grades, Ti-4.5Al-1V-3Fe and Ti-6Al-1V-3Fe, alloys) could be used to predict their flow stress at strain rates up to 1000/s. First, we established from Figures 2 and 3 that when the low strain rate constitutive parameters were used as input codes in DEFORM-3D for uniaxial compression test simulation, the peak flow stress values were comparable with the experimentally derived peak stress. Previous authors reported similar results [24–26], where the flow stress obtained at low strain rates from DEFORM-3D simulations matched the experimental values. The largest difference in the peak stress values was observed at the highest strain rate of 10/s, but this was expected since simulations may not fully represent all the experimental conditions that influence the results. For example, in this study, the friction coefficient was kept constant at 0.3, but under experimental conditions, it changes with temperature, strain, and strain rate [27,28]. Hence, slight deviations are expected between the experimentally determined

peak stress and the DEFORM-3D simulation peak stress. A similar observation was noted in previous studies [24–26]. The maximum permissible discrepancy between experimental and simulation results for hot deformation studies has not been clearly defined. However, from previous studies [24–28], discrepancies of 15.7%, 16%, 22%, and up to 25% have been reported.

**Table 4.** Typical hot working parameters of some metalworking processes [29–31].

| Processes | Strain | Strain Rates (1/s) |
|---|---|---|
| Extrusion | 2.0–5.0 | 0.1–100 |
| Forging/rolling | 0.10–0.50 | 1–1000 |
| Sheet metal forming | 0.10–0.50 | 0.0001–100 |
| Machining | 1.0–10 | 1000–1,000,000 |

Despite the noted difference in the peak stress value at 10/s, the peak flow stress obtained at higher strain rates (20, 50, and 1000/s) fit perfectly on a linear plot with a high correlation value of at least 0.98 (Figure 3). This clearly indicates that the constitutive parameters could be used to generate flow stress data beyond the experimental strain rate.

To validate the peak stress obtained at high strain rates, selected uniaxial compression testing was performed on the commercial-grade Ti-6Al-4V alloy at high strain rates using the Gleeble 3500 thermomechanical simulator. The Ti-6Al-4V alloy was intentionally selected because it is the most studied and well-understood titanium alloy. The results (Figure 5) were compared with the flow stress data generated from the DEFORM-3D uniaxial compression testing simulation (Figure 4). A continuous flow softening trend was observed in both experimental and DEFORM simulation flow curves. However, flow oscillations accompanied flow softening for the experimental condition. This then raises the question of whether the flow oscillation is a metallurgical phenomenon or caused by the load cells of the machine. One major limitation of high strain rate testing using servo-hydraulic systems is the vibration of the load cells, which affects the flow stress values. The corresponding optical microstructure for the two validation experiments showed refined grains with globular or near-equiaxed morphology, which are features that support the oscillations and continuous flow softening seen on the experimental flow curves.

Despite the observed metallurgical features, the authors were unable to quantify how much of the flow oscillation was due to a metallurgical phenomenon and how much could be attributed to the vibration of the servo-hydraulic system. This makes testing at high strain rates using this system challenging from a research point of view. It is worth mentioning that the two experimental results presented in this study to validate the DEFORM-3D simulation were repeated at least three times before the targeted strain rate was achieved. This was because different parametric adjustments had to be made based on trial and error before achieving the desired strain rate. For studies involving the development of new and expensive alloys, such efforts are not beneficial for research and development. Therefore, it is important that a new approach, such as the one presented in this study, be considered, especially where only screw-driven or servo-hydraulic facilities are available.

Based on this study, we found that the peak flow stress values obtained from simulation and experiment at 20 and 50/s strain rates are close (Table 3). The difference in the flow stress values was less than the 20% reported in previous studies [24–26,32,33]. This confirmed that for uniaxial compression testing involving high strain rates, DEFORM-3D simulation could be used in as much as constitutive parameters can be established from experimental data at lower strain rates. The major advantage of this approach is that wear and tear of machine parts, wasting research time and resources, and difficulty reproducing results during high strain rate experimental testing could be avoided.

The limitation of using this approach is that post-deformation characterization of experimental samples will be impossible if only DEFORM-3D is used. Therefore, correlating

microstructural phenomena with the flow curves will not be possible. A few tests must be carried out to validate the simulation results. Future work would consider subjecting other metallic alloys with established constitutive parameters to similar investigations to ascertain the robustness of the approach. Additionally, experimental validation of flow stress obtained at greater than 100/s strain rates using an SHPB should be considered. Since the accuracy of the DEFORM-3D simulation depends on the constitutive models' accuracy, artificial neural networks and other machine learning techniques [34–38] could be incorporated to establish more accurate constitutive parameters that will be fed into the DEFORM-3D software.

## 5. Conclusions

The possibility and reliability of using established constitutive constants determined from low strain rate experimental testing as input codes in DEFORM-3D to generate flow stress at high strain rates were investigated. The following conclusions were drawn from the study:

(a) DEFORM-3D simulation could be used to conduct uniaxial compression test simulations at high strain rates by using established constitutive parameters from low strain rate uniaxial compression testing experiments. The results showed that the relationship between the DEFORM-3D-generated peak flow stress and all strain rates was linear, with a correlation of at least 0.98.

(b) The peak flow stress values predicted by DEFORM-3D simulation were close to those obtained from experimental testing, but a difference of up to 20% may be expected. This difference can be minimized by improving the constitutive models from which the constitutive constants are derived.

**Author Contributions:** Conceptualization, M.B., J.O. and D.K.; methodology, M.B., J.O. and D.K.; software, M.B., J.O. and D.K.; validation, M.B., J.O. and D.K.; investigation, M.B., J.O. and D.K.; writing—original draft preparation, M.B.; writing—review and editing, M.B., J.O. and D.K.; project administration, M.B.; funding acquisition, M.B. All authors have read and agreed to the published version of the manuscript.

**Funding:** This work was supported through the AESA-RISE Fellowship Programme [ARPDF 18-03], the African Materials Science and Engineering Network (the Carnegie-IAS RISE network), and the DST-NRF Centre of Excellence in Strong Materials. AESA-RISE is an independent funding scheme of the African Academy of Sciences (AAS) implemented with the support of the Carnegie Corporation of New York. At the AAS, AESA-RISE is implemented through AESA, the academy's agenda and programmatic platform, created in collaboration with the African Union Development Agency (AUDA-NEPAD). The views expressed in this publication are those of the author(s) and not necessarily those of the AAS, AUDA-NEPAD, or the Carnegie Corporation.

**Institutional Review Board Statement:** Not applicable.

**Informed Consent Statement:** Not applicable.

**Data Availability Statement:** The raw data and supplementary data associated with this study will be made available on request to readers.

**Conflicts of Interest:** The authors declare no conflict of interest. The funders had no role in the design of the study; in the collection, analyses, or interpretation of data; in the writing of the manuscript; or in the decision to publish the results.

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
