# Peer review of "On the Uniaxial Compression Testing of Metallic Alloys at High Strain Rates: An Assessment of DEFORM-3D Simulation"

_applsci, doi:10.3390/app13042686_

Round 1
Reviewer 1 Report
The manuscript is devoted to the DEFORM-3D simulation as an alternative approach to experimental high strain rate uniaxial compression testing. The presented results would be interesting to materials communities. Thus, the manuscript can be recommended to be published in Applied Sciences after the authors addressing the following comments/concerns.
Introduction
reference to the source [11] is missing in the text.
Materials and Methods
Line 116 - "FEM" must be specified in brackets?
Line 128 – “[ref]” - missing reference?
Lines 132-133 (and further in the text) – “s/”, “/s” - units??
Line 146 – “deimension”, “lenth”?
Line 158 – “micscopy”?
Results
Line 179 – Figure 2 - it is necessary to decipher the signatures in Figures 2 (a, b, c). "e" = experiment, "s" - simulation?
Line 181 - perhaps it would be better to sign the axes with letter symbols (ln strain rate looks incomprehensible). It is worth signing legends on figures a, b, c in the same style.
Line 181 – “form”?
Line 194 – “is” presented = are presented.
Line 201 – missing point after “method”?
Line 203 - Perhaps, in the Materials and Methods section, it is worth briefly describing the structure of the studied materials, indicating the grain sizes?
Line 218 – This! trends.
Line 219 – give the decoding of the abbreviation "SHPB" at the first use.
Line 219 – obeained?
Line 224 – “wit”?
Discussion
Line 227 – units?
Line 239 – show = shows.
General comment – it is necessary to proofread the text (many grammatical and spelling errors).
However, these comments do not reduce the relevance and importance of the results. The article is sufficiently novel and interesting to warrant publication. Results and discussion are reliable. References reflect the main publications on which the work is based.
Author Response
Response to reviewer’s comments
The authors appreciated the Editor for considering our manuscript for publication. We also appreciate the reviewers for the useful feedback. We have now revised the manuscript with the hope that the revised version has improved to an acceptable standard. We provide a line by line to each of the comments raised.
Reviewers Comment 1
The manuscript is devoted to the DEFORM-3D simulation as an alternative approach to experimental high strain rate uniaxial compression testing. The presented results would be interesting to materials communities. Thus, the manuscript can be recommended to be published in Applied Sciences after the authors addressing the following comments/concerns.
Introduction
reference to the source [11] is missing in the text.
Response – The authors appreciate the reviewer for picking this up. Reference 11 has been included.
Materials and Methods
Line 116 - "FEM" must be specified in brackets?
Response – FEM has been specified in bracket.
Line 128 – “[ref]” - missing reference?
Response – The reference has been included.
Lines 132-133 (and further in the text) – “s/”, “/s” - units??
Response- This correction has been effected.
Line 146 – “deimension”, “lenth”?
Response- This correction has been effected.
Line 158 – “micscopy”?
Response- This correction has been effected.
Results
Line 179 – Figure 2 - it is necessary to decipher the signatures in Figures 2 (a, b, c). "e" = experiment, "s" - simulation?
Response- This correction has been effected.
Line 181 - perhaps it would be better to sign the axes with letter symbols (ln strain rate looks incomprehensible). It is worth signing legends on figures a, b, c in the same style.
Response- This correction has been effected.
Line 181 – “form”?
Response- This correction has been effected.
Line 194 – “is” presented = are presented.
Response- This correction has been effected.
Line 201 – missing point after “method”?
Response- This correction has been effected.
Line 203 - Perhaps, in the Materials and Methods section, it is worth briefly describing the structure of the studied materials, indicating the grain sizes?
Response- The initial microstructure of the Ti-6Al-4V alloy has been described in the Materials and Methods.
Line 218 – This! trends.
Response- This correction has been effected.
Line 219 – give the decoding of the abbreviation "SHPB" at the first use.
Response- This was given in the introduction.
Line 219 – obeained?
Response- This correction has been effected.
Line 224 – “wit”?
Response- This correction has been effected.
Discussion
Line 227 – units?
Response- The units have been included.
Line 239 – show = shows.
Response- This correction has been effected.
General comment – it is necessary to proofread the text (many grammatical and spelling errors).
Response – The authors appreciate this comment. The entire manuscript was thoroughly revised.
However, these comments do not reduce the relevance and importance of the results. The article is sufficiently novel and interesting to warrant publication. Results and discussion are reliable. References reflect the main publications on which the work is based.
Response – The authors appreciate this comment.

Reviewer 2 Report
Comments article “On the uniaxial compression testing of metallic alloys at high strain rates: an assessment of DEFORM-3D simulation”
As a communication paper, the authors presented the validating of DEFORM-3D simulation of Ti alloys by Gleeble 3500 experimental results, reaching a good accuracy of results and evaluating new alloys, evidencing alternatives to the widely studied and used Ti6Al4V. This paper contributes to the academic community by filling a gap in the literature that was Ti4.5Al1V3Fe and Ti6Al1V3Fe high strain rate data. Furthermore, the DEFORM-3D model was adjusted, and the constitutive parameters were selected to obtain results close to the experimental data, reaching high accuracy and validating the simulation.
The introduction presents the importance of studying and understanding the flow stress mechanisms and how they can help the industry to work with new materials and alloys. The text indicates how simulation saves time and costs compared to experimental testing and evaluations. The Experimental Procedure lists the optimized constitutive parameters for Ti alloys simulations and Gleeble 3500 setup.
The Results and Discussions present graphs that indicate the accuracy and proximity of the experimental and simulation results. The authors compared the advances in using the Gleeble 3500 system and how this equipment helps the researchers to obtain flow stress curves under higher strain rates, which is not feasible by other equipment. Besides that, limitations of the system and future works were proposed, considering artificial neural network and machine learning. Finally, the comparison between simulation data and experimental results was done, indicating the accuracy of the model proposed and the adequate selection of parameters and coefficients. In the end, the authors understood that low-strain constitutive parameters could be successfully applied to high-strain rate simulations.
References
15/34 (44%) are newer than 5 years old. The authors should improve the number of recent publications on the theme, which should indicate an actual interest of the academic community in the theme studied in this article.
Some punctual revisions:
Line 92: the ratio is expressed in “s–1”, but in Line 132 it is expressed in “s/”, and in other parts of the text it is shown “/s”. The authors should review the units in the whole manuscript and adopt a standard one.
Line 128: There is an error on the reference, [ref]. The authors should check what is the missing reference.
Line 138: Table 2: “Coefficient of heat convention”, is this parameter correctly named? Or is it heat convection?
Line 143: “Gleeble 3500”, insert (city, country) of manufacturer.
Line 179: Figure 2: The Y-axis label is “Stress”. The authors should review it in (a) where the Y-axis is named “Flow Stress”.
Line 179: Figure 2: The X-axis should have the same range for all the materials, (a), (b), and (c). The values seen in (c) 780~960 are the best ones, and the authors should replicate them in (a) and (b).
Line 179: Figure 2: The results present experimental points for different temperatures, 800, 850, 900, and 950°C; however, in the Materials and Methods section, the authors presented the experiments conducted in a single temperature, 900°C (Line 146-150). The authors should unify the information or explain why they are different.
Line 191: The authors should review a typing error: “confrim” should be replaced by “confirm”.
Line 206: The authors should review a typing error: “recrustallisation” should be replaced by “recrystallisation”.
Line 211: Figure 4: The curves represent Gleeble experiments conducted at 800°C and 900°C, but in the Materials and Methods section, the authors present the Gleeble testing performed at 900°C (Line 146-150), and in Figure 2, the authors present experimental data collected at 800, 850, 900, and 950°C. The authors should comment the differences or unify the information.
Line 211: Figure 4: The X-axis label in (e) and (f) is “Temperature”. The authors should review and correct it to “Strain”.
Line 223: Figure 5: The curves (a) and (b) should be merged in one only figure, labelling them.
Line 265: The authors should reference the affirmation “… in this study the friction coefficient was kept constant at 0.3, but under experimental condition, it actually changes with temperature, strain and strain rate.”. Literature recommended: doi.org/10.1007/s12289-008-0170-5; doi.org/10.1088/1742-6596/1378/3/032094.
Author Response
Response to reviewer’s comments
The authors appreciated the Editor for considering our manuscript for publication. We also appreciate the reviewers for the useful feedback. We have now revised the manuscript with the hope that the revised version has improved to an acceptable standard. We provide a line by line to each of the comments raised.
Reviewers Comments 2
As a communication paper, the authors presented the validating of DEFORM-3D simulation of Ti alloys by Gleeble 3500 experimental results, reaching a good accuracy of results and evaluating new alloys, evidencing alternatives to the widely studied and used Ti6Al4V. This paper contributes to the academic community by filling a gap in the literature that was Ti4.5Al1V3Fe and Ti6Al1V3Fe high strain rate data. Furthermore, the DEFORM-3D model was adjusted, and the constitutive parameters were selected to obtain results close to the experimental data, reaching high accuracy and validating the simulation.
The introduction presents the importance of studying and understanding the flow stress mechanisms and how they can help the industry to work with new materials and alloys. The text indicates how simulation saves time and costs compared to experimental testing and evaluations. The Experimental Procedure lists the optimized constitutive parameters for Ti alloys simulations and Gleeble 3500 setup.
The Results and Discussions present graphs that indicate the accuracy and proximity of the experimental and simulation results. The authors compared the advances in using the Gleeble 3500 system and how this equipment helps the researchers to obtain flow stress curves under higher strain rates, which is not feasible by other equipment. Besides that, limitations of the system and future works were proposed, considering artificial neural network and machine learning. Finally, the comparison between simulation data and experimental results was done, indicating the accuracy of the model proposed and the adequate selection of parameters and coefficients. In the end, the authors understood that low-strain constitutive parameters could be successfully applied to high-strain rate simulations.
References
15/34 (44%) are newer than 5 years old. The authors should improve the number of recent publications on the theme, which should indicate an actual interest of the academic community in the theme studied in this article.
Response – The authors appreciate this comment. We have added a few more recent articles as suggested. However, the references we have cited that are more than 5 years old were mostly primary authors that made the initial contributions prior to other researchers. I hope this manuscript will not be penalised should we not improve the reference list beyond its current state.
Some punctual revisions:
Line 92: the ratio is expressed in “s–1”, but in Line 132 it is expressed in “s/”, and in other parts of the text it is shown “/s”. The authors should review the units in the whole manuscript and adopt a standard one.
Response – The authors appreciate this comment. The unit is now uniform in the entire manuscript.
Line 128: There is an error on the reference, [ref]. The authors should check what the missing reference is.
Response – The authors appreciate this comment. The reference has been included.
Line 138: Table 2: “Coefficient of heat convention”, is this parameter correctly named? Or is it heat convection?
Response – The authors appreciate this comment. Convention has been changed to convection.
Line 143: “Gleeble 3500”, insert (city, country) of manufacturer.
Response – The manufacturer details have been included. Thanks to the reviewer for this comment.
Line 179: Figure 2: The Y-axis label is “Stress”. The authors should review it in (a) where the Y-axis is named “Flow Stress”.
Response – The authors appreciate this comment. The Y- axis title has been changed to peak stress in Figure 2.
Line 179: Figure 2: The X-axis should have the same range for all the materials, (a), (b), and (c). The values seen in (c) 780~960 are the best ones, and the authors should replicate them in (a) and (b).
Response – The authors appreciate this comment. The scale in Figure 2 has been standardised.
Line 179: Figure 2: The results present experimental points for different temperatures, 800, 850, 900, and 950°C; however, in the Materials and Methods section, the authors presented the experiments conducted in a single temperature, 900°C (Line 146-150). The authors should unify the information or explain why they are different.
Response – The authors appreciate this comment. This has been addressed in section 2.1 of the revised manuscript. See extract below:
The DEFORM-3D simulation results were validated for low and high strain rate conditions. For the low strain rates (0.01-10/s), the peak stress from published flow stress data of three titanium (α+β) alloys – Ti-4.5Al-1V-3Fe, Ti-6Al-1V-3Fe and Ti-6Al-4V were used [16-19]. The data were generated from uniaxial compression testing on Gleeble 3500 thermomechanical simulator at deformation temperatures of 800-950°C. For the high strain rate test, uniaxial compression testing was conducted on commercial grade Ti-6Al-4V alloy with a complex initial microstructure. The complex microstructure consisted of nearly equiaxed alpha grains (diameter = 5 ± 2 μm), elongated alpha grains (length is between 12 and 49 μm) and a network of intergranular beta phase.
Line 191: The authors should review a typing error: “confrim” should be replaced by “confirm”.
Response – The authors appreciate this comment. The correction has been made.
Line 206: The authors should review a typing error: “recrustallisation” should be replaced by “recrystallisation”.
Response – The authors appreciate this comment. The correction has been made.
Line 211: Figure 4: The curves represent Gleeble experiments conducted at 800°C and 900°C, but in the Materials and Methods section, the authors present the Gleeble testing performed at 900°C (Line 146-150), and in Figure 2, the authors present experimental data collected at 800, 850, 900, and 950°C. The authors should comment the differences or unify the information.
Response – The authors appreciate this comment. This has been addressed in section 2.1 of the revised manuscript. See extract below:
The DEFORM-3D simulation results were validated for low and high strain rate conditions. For the low strain rates (0.01-10/s), the peak stress from published flow stress data of three titanium (α+β) alloys – Ti-4.5Al-1V-3Fe, Ti-6Al-1V-3Fe and Ti-6Al-4V were used [16-19]. The data were generated from uniaxial compression testing on Gleeble 3500 thermomechanical simulator at deformation temperatures of 800-950°C. For the high strain rate test, uniaxial compression testing was conducted on commercial grade Ti-6Al-4V alloy with a complex initial microstructure. The complex microstructure consisted of nearly equiaxed alpha grains (diameter = 5 ± 2 μm), elongated alpha grains (length is between 12 and 49 μm) and a network of intergranular beta phase.
Line 211: Figure 4: The X-axis label in (e) and (f) is “Temperature”. The authors should review and correct it to “Strain”.
Response – The authors appreciate this comment. Figure 4 has been revised.
Line 223: Figure 5: The curves (a) and (b) should be merged in one only figure, labelling them.
Response – The authors appreciate this comment. Figure 5 has been revised.
Line 265: The authors should reference the affirmation “… in this study the friction coefficient was kept constant at 0.3, but under experimental condition, it actually changes with temperature, strain and strain rate.”. Literature recommended: doi.org/10.1007/s12289-008-0170-5; doi.org/10.1088/1742-6596/1378/3/032094.
Response – The authors appreciate the articles recommended. We have cited them accordingly.

Round 2
Reviewer 1 Report
In general, the main comments from the first review were corrected by the authors. In the current version, the manuscript can be recommended for publication in the journal Applied Sciences.